# Cutaneous Manifestations in SARS-CoV-2 Infection—A Series of Cases from the Largest Infectious Diseases Hospital in Western Romania

**DOI:** 10.3390/healthcare9070800

**Published:** 2021-06-25

**Authors:** Ruxandra Laza, Virgil Filaret Musta, Narcisa Daniela Nicolescu, Adelina Raluca Marinescu, Alexandra Mocanu, Laura Vilceanu, Roxana Paczeyka, Talida Georgiana Cut, Voichita Elena Lazureanu

**Affiliations:** 1Department XIII, Discipline of Infectious Diseases, University of Medicine and Pharmacy “Victor Babes” Timisoara, E. Murgu Square, Nr. 2, 300041 Timisoara, Romania; ruxi_martincu@yahoo.com (R.L.); musta.virgil@umft.ro (V.F.M.); nicolescu.narcisa@umft.ro (N.D.N.); talida.cut@umft.ro (T.G.C.); lazureanu.voichita@umft.ro (V.E.L.); 2Clinical Hospital of Infectious Diseases and Pneumophtisiology “Doctor Victor Babes” Timisoara, Gheorghe Adam, Nr. 13, 300310 Timisoara, Romania; alexandramocanu021@yahoo.com (A.M.); vilceanulaura10@gmail.com (L.V.); roxy_klopo@yahoo.com (R.P.); 3Doctoral School, University of Medicine and Pharmacy “Victor Babes” Timisoara, E. Murgu Square, Nr. 2, 300041 Timisoara, Romania

**Keywords:** SARS-CoV-2, cutaneous manifestations, erythematous-macular rash, immune response

## Abstract

(1) Background: SARS-CoV-2 infection, which appeared as an isolated epidemic outbreak in December 2019, proved to be so contagious that, within 3 months, the WHO declared COVID-19 a pandemic. For one year (pre-vaccination period), the virus acted unhindered and was highly contagious, with a predominantly respiratory-oriented aggression. Although this lung damage, responsible for the more than 3,090,025 deaths, has provided sufficient data to facilitate the understanding of pathogenic mechanisms, other observation data, which meet the quality of emerging clinical aspects, such as rashes, remain without well-defined etiopathogenic support or a well-contoured clinical framework. (2) Methods and Results: We followed the occurrence of cutaneous manifestations in patients hospitalized during the second and third outbreak of SARS-CoV-2 in the main clinics of infectious diseases of our county, Timis, and recorded laboratory investigations and clinical evolution for five suggestive cases. (3) Conclusions: The presented cases, added to many other present and future clinical observations, will allow for better knowledge and understanding of SARS-CoV-2 infection, a requirement that has become a global priority for the entire medical and scientific community.

## 1. Introduction

The presence of coronaviruses in human pathology offers us a model of natural evolution of the relationship between the living world of microorganisms and humanity, that is, between virus and human [1]. Since they were first isolated (1965), they have behaved as infectious agents with low pathogenicity, causing seasonal cough or benign enteritis. Gradually, under the action of mutational factors related to microbial ecology, the environment, and successive transitions in various populations, they acquired new qualities and epidemiologic strains developed [2].

Their new qualities of contagiousness, invasiveness, and pathogenicity have ensured their ability to trigger severe acute respiratory syndrome (SARS-CoV-1), which concluded in 2002/2003 with 8000 cases and 774 deaths in over 26 countries and 5 continents [3]. Ten years after SARS-CoV-1, another highly pathogenic coronavirus, Middle East respiratory syndrome coronavirus (MERS-CoV), emerged in the Arabian Peninsula, with a case fatality rate above 40% [4,5]. The year 2019 added extra vigor to the pathogenic properties of β-coronavirus genus. The SARS-CoV-2 epidemic outbreak originally developed in Wuhan, China, and within three months, it was declared a global pandemic by the World Health Organization [6]. Unfavorable developments and complications became further causes for concern [7], and a great diversity of clinical presentations and that children appeared relatively immune were noted [8].

It has been proven that SARS-CoV-2 primarily targets the upper respiratory mucosa using angiotensin-converting enzyme 2 (ACE2) as a functional receptor for the viral spikes. We know that in the pathogenesis of SARS-CoV-2 infection, a first step is performed by the spike viral membrane protein, which attaches to the host cell via the angiotensin-converting enzyme 2 (ACE2); from that moment, under the action of the transmembrane protease serine 2 (TMPRSS2), the fusion of the viral envelope with the cell membrane is triggered. The entry of the virus into the cell is followed by the release of its RNA-like genome. Expression of the ACE2 receptor for the spike protein of SARS-CoV-2 has been demonstrated in a number of human cells, including keratinocytes and adipocytes, thus suggesting a possible susceptibility to disruption of the skin barrier [9,10,11,12]. 

The presence of cutaneous manifestations as a new emergent clinical element [13] motivated Galván Casas et al. to publish, based on their clinical expertise and the available literature, the first COVID-19-related skin manifestations classification. Based on their findings, COVID-19-associated cutaneous lesions can be divided into five morphological patterns: livedo or necrosis (6%), vesicular eruptions (9%), pseudo-chilblains (19%), urticarial lesions (19%), and maculopapular lesions (47%) [14]. However, there are still insufficient data supporting the relationship between cutaneous manifestations and the severity of the respiratory condition [15].

## 2. Materials and Methods

This is a retrospective case series describing the evolution of 5 out of 4371 cases of patients hospitalized in the clinics of the Hospital for Infectious Diseases and Pneumophtisiology “Victor Babes” Timisoara during the second and third outbreaks of COVID-19, from 1 August 2020 to 1 April 2021. Clinical characteristics, personal history, and laboratory and imagistic results were recorded, and the patients’ evolution and outcome were followed.

## 3. Results

### 3.1. Case I

A three-year-old Caucasian girl presented to the territorial Pediatric Emergency Department complaining of an afebrile sudden onset of an erythematous eruption, disposed in extensive plaques on the torso and limbs. The patient was otherwise healthy with no comorbidities, medications, or previous drug reactions. She was evaluated, and naso-pharyngeal swabs testing for SARS-CoV-2 RNA amplification were positive. Admission into the Infectious Diseases Clinic for further investigations and close observation was suggested but denied by the child’s legal guardian; thus, antihistaminic and calcium treatment was prescribed. Over the following days, she presented mild fever episodes with a maximum of 38.5 °C body temperature but, otherwise, was in a generally good condition. The recurrent fever episodes in association with persistent erythematous macules, despite antihistaminic treatment that was applied, led to her admission, after 7 days of illness, to our Infectious Diseases Department, as shown in Figure 1.

In the first two days of admission, she presented isolated febrile episodes, which remitted after administration of oral antipyretics. During the five-day hospitalization period, her general condition remained relatively good. Laboratory investigations revealed normal white blood cell count, normal renal and liver functions, increased D-dimers, and minimal nonspecific inflammation. Extensive laboratory results are presented in Table 1. Starting with the eighth day from symptoms onset, the rash tended to fade on the trunk, but acral target and targetoid lesions persisted throughout the hospitalization period. Complete resolution of erythema was observed after 13 days from onset.

### 3.2. Case II

A 53-year-old Caucasian woman, with a known history of hypercholesterolemia and osteoporosis, presented to the Emergency Department complaining of nasal obstructions, severe asthenia, irritating dry cough, and worsening shortness of breath that had started 5 days before. SARS CoV-2 RNA amplification from naso-pharyngeal swabs tested positive, and the patient was admitted to the Infectious Diseases Department. Laboratory investigations revealed a normal WBC and RBC count, elevated D-dimer levels (0.97 μg/mL), and minimal increased C-reactive protein (9.33 mg/L). Central and peripheral focal areas with pseudo-nodular appearance of ground glass were detected on the CT scan. The patient started off-label antiviral therapy, following Romanian Ministry of Health guidelines. In the subsequent days, she promptly recovered and was discharged on demand, clinically improved after a nine-day period of hospitalization. Four days from hospital discharge, she became feverish with a generalized erythematous, intensely pruritic rash. The onset of the erythematous lesions coincided with an antibiotic course of amoxicillin (500 mg PO, q8h), prescribed by a dentist for recurrent pulpitis. First spotted on the face, the urticarial lesions quickly spread to the trunk and limbs with a tendency to become confluent as presented in Figure 2. The patient was readmitted to our clinic. Laboratory investigations, as detailed in Table 1, revealed a normal blood count and increasing C-reactive protein and ferritin. Antihistamine, corticosteroid, and calcium treatment was instituted. The patient became afebrile on the second day of hospitalization, but urticarial lesions persisted. Upon discharge, the rash greatly diminished in intensity and pruritus. From the patient’s accounts, it completely disappeared a week after the second discharge (approximately 14 days after the onset).

### 3.3. Case III

A 59-year-old Caucasian woman, with a known personal history of autoimmune thyroiditis under treatment with levothyroxine, was admitted to the Infectious Diseases Department, due to dysphagia, chills, dry cough, and headaches. Naso-pharyngeal swabs tested for SARS-CoV-2 resulted positive, and native CT scan described several areas of peripheral ground-glass affecting less than 10% of the lung surface, that is, a mild form of SARS-CoV-2 pneumonia. Laboratory test results, detailed in Table 2, emphasized leukopenia with lymphopenia and neutropenia (remitted within 4 days of hospitalization), no biological inflammatory syndrome, and D-dimers within normal limits. Complex treatment followed, with a favorable evolution. The patient was discharged with a negative SARS-CoV-2 RNA amplification test after 16 days of hospital stay. Seven days after the hospital discharge, on the 27th day from onset of symptoms, small papular erythematous lesions with particular face and acral involvement, as presented in Figure 3, appeared. In the following days, gradual spontaneous improvement of the skin lesions was noted. 

### 3.4. Case IV

A 46-year-old Caucasian woman, otherwise healthy with no comorbidities, allergic to penicillin and trimethoprim/sulfamethoxazole, was admitted to the hospital due to fever, myalgias, arthralgias, headaches, dysphagia, upper abdominal pain, and dry cough. At the time of hospitalization, the patient had been ill for about four days, and naso-pharyngeal swabs tested positive for SARS-CoV-2 infection. The native CT scan described a mild form of COVID-19 pneumonia with minimum bilateral subpleural focal areas of homogeneous ground-glass. The laboratory test results, as detailed in Table 2, presented values within normal limits. On the eighth day of hospitalization, the patient continued to present persistent dry cough, chills, low-grade fever, myalgias, sweating, and marked asthenia. Upon general examination, urticarial lesions similar to hives located on the patient’s face, neck, and torso, as presented in Figure 4, were observed. An increase in the WBC count as well as in the C-reactive protein and fibrinogen levels was noted (Table 2). A second round of chest computer tomography was performed, and the results revealed moderately progressive areas of pulmonary condensation (ground-glass and crazy-paving type), affecting up to 50% of the pulmonary area. The therapy was supplemented with remdesivir, antihistamines, and calcium tablets. Gradual improvement of the rash was noted within 4–5 days. Seven days following their onset, total resolution of the urticarial lesions was noted and the patient was discharged from the hospital on demand. 

### 3.5. Case V

A 65-year-old Caucasian woman, with a known personal history of hypertension and megacolon, presented to the Emergency Department complaining of dysphagia, low-grade fever, productive cough, sweating, chills, headaches, and asthenia, which had started eight days before. Naso-pharyngeal swabs tested positive for SARS-CoV-2 RNA amplification, and hospital admission was recommended but refused by the patient. Due to a period of 19 days’ sustained symptoms from onset, the patient was admitted for further investigations to the Infectious Diseases Department. 

On admission, the patient presented with affected general condition, marked asthenia, interscapulovertebral burning pain, productive cough, dysphagia, sweating, and rosacea-like rash on acral sites (Figure 5). The CT scan documented a mild form of COVID-19 pneumonia with discrete ground-glass areas corresponding to the apical segment of the lower left lobe. Laboratory tests, as detailed in Table 3, detected lymphopenia and a slight increase in glucose blood levels. Antiviral therapy and subcutaneous low-molecular-weight heparin following SARS-CoV-2 treatment guidelines was initiated. In the subsequent days, the patient slowly recovered and was discharged with a discrete erythematous rash on her upper torso. 

## 4. Discussion

Cutaneous manifestations, as observed in our case study, may represent the single feature of COVID-19 or they may suggest a relapse in the evolution of the disease or may occur in the immediate recovery period.

Throughout the pandemic, the pediatric population have remained relatively unscathed from severe COVID-19-related complications [16]. The first clinical case we presented (the three-year-old patient) is characterized by the fact that the SARS-CoV-2 infection had, as its only and main clinical expression, an erythema multiforme-like rash consisting of target and targetoid lesions on the trunk and limbs, accompanied by fever from the second to the eighth day. The lesions persisted for 13 days against the background of a slightly influenced general condition (the girl was adynamic, with no interest in play), and SARS-CoV-2 RNA amplification tested positive on the 10th day of illness. Slightly elevated D-dimer values (1.4 μg/mL) remained without clinical support. These observations corelate with the findings of Torrelo et al., which state the fact that an erythema multiforme-like rash may commonly appear in children and may be associated with a mild COVID-19 course [9,17]. 

According to Galván Casas et al., maculopapular eruptions and urticarial lesions are the most frequently encountered cutaneous manifestations and present similar patterns of associated findings. Pruritus occurred in 92% of cases of urticarial lesions and in 56% of cases of maculopapular lesions [13]. Approximately 55.8% of cases of rashes containing macules and papules, reported by Tan et al., occurred during the active phase of the disease [18]. In the second clinical case (the 53-year-old patient), the rash appeared on the 21st day of the disease and was of erythematous and intense pruritic type. The erythematous papular lesions, with a tendency to confluence, covered the face, trunk, and limbs. It lasted 14 days and was associated with fever in the first two days, and the patient otherwise maintained an overall general good condition. The onset of the rash coincided with the first day of antibiotic treatment (amoxicillin 3 × 500 mg), prescribed by a dentist. The absence of an anamnestic atopic terrain and of accompanying systemic reactions at the onset of the rash along with the presence of a generalized macular exanthema and erythematous papular skin response, with a tendency to confluence, that persisted even after the discontinuation of antibiotic, prompted us to maintain the association of rash with the underlying disease, and especially the involvement of small and medium vessels in the dermis and hypodermis as part of an immunopathological response. 

The third clinical case is characterized by an acral ovulary-shaped erythematous papular rash on the 27th day of the disease in accordance with Freeman et al.’s proposed severity of COVID-19 scale [19]. Reported lymphopenia did not significantly affect the negation of the SARS-CoV-2 RNA amplification test (recorded on the 17th day of illness). The etiopathogenic mechanism of the rash highlights the involvement of one of the immune response links targeting the endothelial cell, capable of producing cytokines and chemotactic factors or stimulating factors of blood cell colonies [20].

Case IV (the 46-year-old patient) records the possibility of a moment of recrudescence settling on the 12th day of illness (8th of hospitalization), with the appearance of a urticarial rash with slightly raised papular erythematous lesions on the face, neck, and torso, accompanied by exacerbation of respiratory and general charges (irritating cough, chills, and sweating). The rash lasted for eight days. According to Marzano et al., urticarial exanthems have been associated with severe COVID-19 [21]. This statement is applicable to our 46-year-old patient, sustained by the progression of lung injuries. The pathogenic mechanism of the rash remains unclear, either as part of a viremia or as part of an immunopathological reaction of the small and medium vessels in the dermis and hypodermis.

Rosacea-like lesions on acral sites (face) or exacerbation of such a preexistent condition is attributed, by Zheng et al., to prolonged use of protective personal items (PPE) used in occupational hazards [22]. As opposed to this statement, in the fifth clinical case, the lesions emerged on the 20th day of illness and were unrelated to facial protection materials. The test for SARS-CoV-2 RNA amplification on the 29th day from symptom onset was negative, and significant lymphopenia was not shown to elicit an inadequate antiviral immune response.

## 5. Conclusions

Although rare, cutaneous manifestations are part of the clinical picture of the COVID-19 disease [23,24]. Our case study shows that cutaneous manifestations may be the only manifestation of the disease, may suggest a moment of recrudescence in the evolution of the disease, or may occur in the immediate convalescence of the disease. Our patients were classified as having moderate forms of COVID-19.

The cutaneous manifestations, under morphological and topographical aspect, were categorized as maculopapular eruptions, urticarial lesions, rosacea-like eruption, and erythema multiforme-like lesions, partially abiding to the classification of Galván Casas et al. 

Skin lesions, as a unique element or one associated with other clinical changes, requires that the etiology of COVID-19 not be omitted from the diagnostic judgment of the practitioner, regardless of the profile of their specialty. Lymphopenia present in two of the patients was associated with a relative delay in COVID-19 test negativity. 

The etiopathogenesis of cutaneous manifestations awaits a response on the molecular substrate level and the establishment of delimitations between the two factors directly involved: the pathogen and the realization of the immune response of the host organism [25,26].

The presented cases, added to numerous other current and future clinical observations, will allow for better knowledge and understanding of SARS-CoV-2 infection, a requirement that has become a global priority for the entire medical and scientific community.

## Figures and Tables

**Figure 1 healthcare-09-00800-f001:**
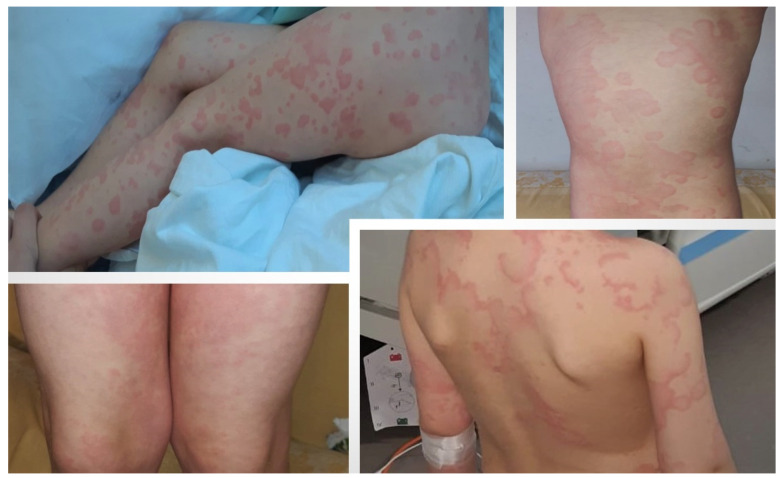
Case I: A three-year-old girl with typical target and targetoid lesions in COVID-19-related erythema multiforme. The first two upper pictures taken upon hospital admission (eighth day of illness), and the base pictures taken on the third day of admission.

**Figure 2 healthcare-09-00800-f002:**
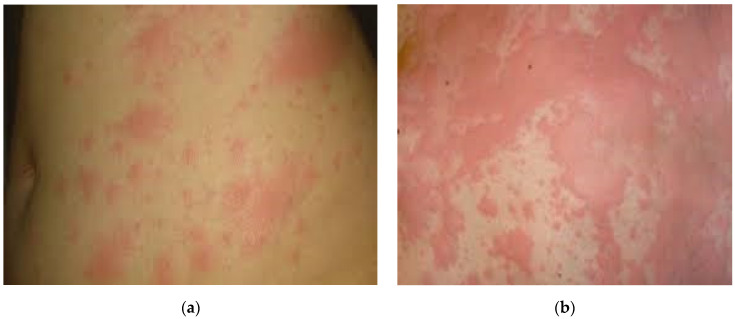
Case II: (**a**) Pruritic disseminated erythematous skin rash initially affecting the face, followed by rapid spreading to the trunk and upper and lower extremities; (**b**) Details of erythematous papular lesions aggregated into confluent larger erythematous patches on the trunk.

**Figure 3 healthcare-09-00800-f003:**
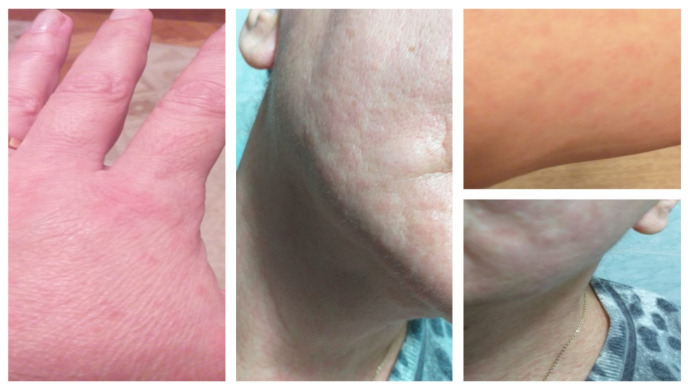
Case III: Multiple small oval erythematous papules with different degrees of confluence and particular face and acral involvement.

**Figure 4 healthcare-09-00800-f004:**
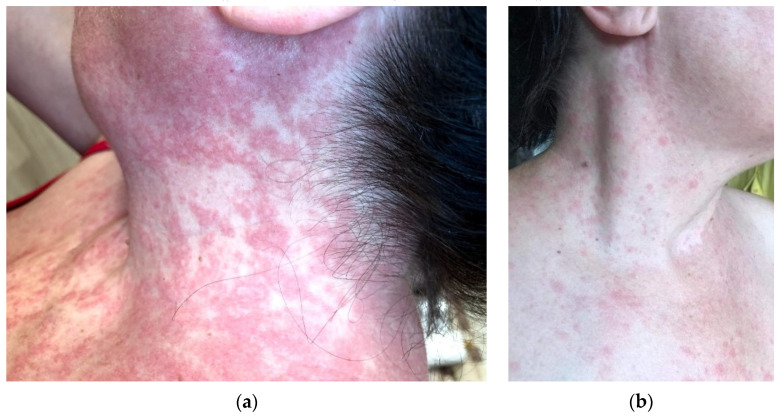
Case IV: (**a**) Urticarial rash with slightly raised papular erythematous lesions, presented on face, neck, and torso; (**b**) Demised urticarial lesions after 5 days of antihistaminic treatment.

**Figure 5 healthcare-09-00800-f005:**
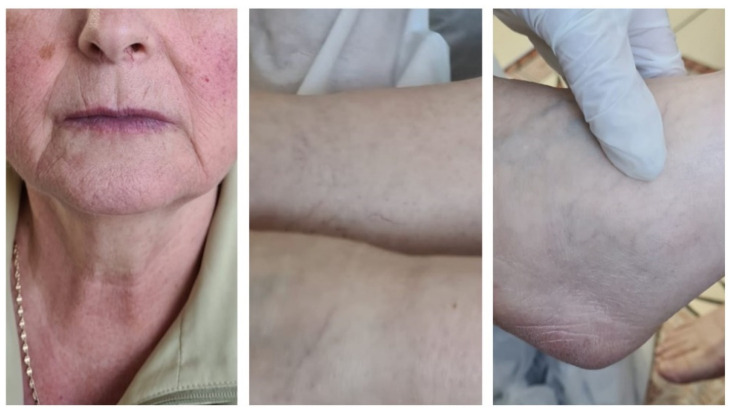
Case V: Rosacea-like rash on acral sites and discrete erythematous nonpruritic rash on neck and torso.

**Table 1 healthcare-09-00800-t001:** Laboratory data of Case I and II.

Laboratory Data	Normal Range	Case I Admission	Case I Follow-Up	Case II Admission	Case II Follow-Up
RBC	4,000,000–5,500,000/mmc	4,120,000	4,470,000	5,090,000	4,290,000
Hemoglobin	11.5–16 g/dL	10.6 ↓	11.5	14	11.9
Hematocrit	35.5–47%	31.5 ↓	34.2 ↓	42.1	36.1
Platelets	150,000–400,000/mmc	338,000	406,000	323,000	214,000
WBC	4000–10,000/mmc	7280	8760	9300	5920
NE	1500–7500 (30–65%)	3790 (52.1%)	2700 (30.8%)	5760 (61.9%)	3530 (59.6%)
LY	3000–9500 (30–55%)	3170 (43.5%)	5060 (57.8%)	2600 (28%)	1580 (26.7%)
Fibrinogen	1.8–3.5 g/L	3.18	2.66	4.62 ↑	4.44 ↑
D-Dimers	<0.5 μg/mL	1.41 ↑	0.97 ↑	0.97 ↑	0.81 ↑
PT	9.8–12.1 s	12.8 ↑	11.7	10	10.1
PT	70–120%	89.6	107.3	>130 ↑	>130 ↑
INR	<1.1	1.05	0.96	0.80	0.80
APTT	22.1–28.1 s	23.6	20.4 ↓	18.1 ↓	19.8 ↓
Lactate	4.5–19.8 mg/dL	16.39	11.74	14.86	12.8
AST	0–32 U/L	48.1 ↑	26.2	21.5	17.2
ALT	0–33 U/L	13.4	8.3	27.8	35.5 ↑
Serum urea	0–50 mg/dL	31.7	24.1	36.3	18.5
Creatinine	0.50–0.90 mg/dL	0.32	0.22	0.64	0.53
Glucose	74–106 mg/dL	129 ↑	74	95	91
Ferritin	15–150 μg/L	42.97	19.13	118.98	235.31 ↑
CRP	0–5 mg/L	12.5 ↑	1.92	9.33 ↑	26.75 ↑
IL-6	0–7 pg/mL	-	<1.50	-	-

Legend: WBC—white blood cells; RBC—red blood cells; NE—neutrophils; LY—lymphocytes; PT—prothrombin time; INR—international normalized ratio; APTT—activated partial thromboplastin time; AST—aspartate aminotransferase; ALT—alanine aminotransferase; CRP—C-reactive protein; IL-6—interleukin 6, elevated values are marked with purple color and ↑, decreased values are marked with red color and ↓.

**Table 2 healthcare-09-00800-t002:** Laboratory data for Case III and IV.

Laboratory Data	Normal Range	Case III Admission	Case III First Follow-Up	Case III Second Follow-Up	Case IV Admission	Case IV First Follow-Up	Case IV Second Follow-Up
RBC	4,000,000–5,500,000/mmc	4,690,000	4,700,000	5,020,000	4,580,000	4,900,000	4,470,000
Hemoglobin	11.5–16 g/dL	14.3	14.3	15.1	14.4	15.1	13.9
Hematocrit	35.5–47%	41.7	41.3	42.7	41.1	43.1	38.7
Platelets	150,000–400,000/mmc	159,000	175,000	252,000	201,000	358,000	241,000
WBC	4000–10,000/mmc	2260 ↓	7130	6840	4700	17,760 ↑	13,870 ↑
NE	1500–7500 (30–65%)	900 (39.9%) ↓	6000 (84.2%)	5660 (82.7%)	2850 (60.6%)	15,340 (6.4%) ↑	8680 (31.4%)
LY	3000–9500 (30–55%)	1150 (50.9%) ↓	830 (11.6%) ↓	880 (12.9%) ↓	1540 (32.8%) ↓	1900 (10.7%) ↓	4360 (31.4%)
Fibrinogen	1.8–3.5 g/L	3.03	3.19	-	3.58 ↑	4.85 ↑	3.39
D-Dimers	<0.5 μg/mL	0.31	0.36	0.4	0.41	0.41	0.34
PT	9.8%–12.1 s	10.9	10.1	-	11.7	10.5	-
PT	70–120%	>130 ↑	>130 ↑	-	112.8	>130 ↑	-
INR	<1.1	0.87	0.80	-	0.94	0.84	-
APTT	22.1–28.1 s	27.6	24.1	-	25.1	21.3 ↓	-
Lactate	4.5–19.8 mg/dL	9.88	41.18 ↑	35.85 ↑	9.87	23.33 ↑	-
AST	0–32 U/L	20.2	60.1 ↑	14.1	27.6	11.9	10.1
ALT	0–33 U/L	18.1	62.7 ↑	33.4 ↑	27.5	21.3	21.5
Serum urea	0–50 mg/dL	23.5	33.1	41.5	19.1	33.4	34.8
Creatinine	0.50–0.90 mg/dL	0.78	0.72	0.67	0.83	0.61	0.54
Glucose	74–106 mg/dL	90	131	139	96	76	247 ↑↑
Ferritin	15–150 μg/L	385.99 ↑	514.8 ↑	2524.4 ↑↑↑	28.14	71.98	145.39
CRP	0–5 mg/L	2.47	1.13	0.58	1.37	6.51 ↑	2.67
IL-6	0–7 pg/mL	12.42 ↑	7.59 ↑	-	11.96 ↑	<1.50	4.47

Legend: WBC—white blood cells; RBC—red blood cells; NE—neutrophils; LY—lymphocytes; PT—prothrombin time; INR—international normalized ratio; APTT—activated partial thromboplastin time; AST—aspartate aminotransferase; ALT—alanine aminotransferase; CRP—C-reactive protein; IL-6—interleukin 6, elevated values are marked in purple color with one ↑ for slightly elevated values, ↑↑ for moderate elevated values, ↑↑↑ for highly increased values, decreased values are marked with red color and ↓.

**Table 3 healthcare-09-00800-t003:** Laboratory data of Case V.

Laboratory Data	Normal Range	Case V Admission	Case V Follow-Up
RBC	4,000,000–5,500,000/mmc	4,600,000	4,350,000
Hemoglobin	11.5–16 g/dL	12.7	12.2
Hematocrit	35.5–47%	38	36
Platelets	150,000–400,000/mmc	279,000	198,000
WBC	4000–10,000/mmc	8210	11,750 ↑
NE	1500–7500 (30–65%)	7350 (89.5%)	9770 (83.2%) ↑
LY	3000–9500 (30–55%)	770 (9.4%) ↓	920 (7.8%) ↓
Fibrinogen	1.8–3.5 g/L	2.70	1.70
D-Dimers	<0.5 μg/mL	0.59	0.37
PT	9.8–12.1 s	11.1	10.7
PT	70–120%	115	125.1
INR	<1.1	0.93	0.90
APTT	22.1–28.1 s	23	20.3
Lactate	4.5–19.8 mg/dL	16.95	20.21 ↑
AST	0–32 U/L	15.1	51.6 ↑
ALT	0–33 U/L	14.6	70.9 ↑
Serum urea	0–50 mg/dL	16.3	53.2 ↑
Creatinine	0.50–0.90 mg/dL	0.61	0.68
Glucose	74–106 mg/dL	141 ↑	115 ↑
Ferritin	15–150 μg/L	37.16	239.81 ↑
CRP	0–5 mg/L	1.24	0.31
IL-6	0–7 pg/mL	<1.5	-

Legend: WBC—white blood cells; RBC—red blood cells; NE—neutrophils; LY—lymphocytes; PT—prothrombin time; INR—international normalized ratio; APTT—activated partial thromboplastin time; AST—aspartate aminotransferase; ALT—alanine aminotransferase; CRP—C-reactive protein; IL-6—interleukin 6, elevated values are marked with purple color and ↑, decreased values are marked with red color and ↓.

## Data Availability

The data presented in this study are openly available at http://dx.doi.org/10.17632/x4j8f2jdmz.1 (accessed on 12 May 2021).

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
