# Peer review of "Cutaneous Manifestations in SARS-CoV-2 Infection—A Series of Cases from the Largest Infectious Diseases Hospital in Western Romania"

_healthcare, 2021, doi:10.3390/healthcare9070800_

Round 1

Reviewer 1 Report

General comments: the manuscript by Laza et al regards the cutaneous manifestations of covid19 from the Hospital for Infectious Diseases and Pneumophtisi- 50 ology Victor Babes Timisoara.

The authors reported for each patient laboratory tests and clinical images, highlighting the most common clinical skin presentations of SARSCov2 such as the maculopapular rash  

Comments: I suggest to clarify the tables (please add normal values in another column because it seems to be a little bit confused) and to add In the title something like “a real life experience from a Romanian Hospital” or something similar just to underline that you are talking about your experience.  

Adding also the possibile etiopatogenesis process could be useful for the readers.

Reviewer 2 Report

This is a nice case series, however, it requires further editing to allow publication. 

Major remarks:

  • The whole manuscript requires editing for English language. There are numerous flaws both in spelling and grammar. Consistently use past tense.
  • The cited literature is insufficient. Not only is the total number of citations too low, the given references are mainly case series from minor journals. The most important work "Classification of the cutaneous manifestations of COVID-19: a rapid prospective nationwide consensus study in Spain with 375 cases" published in Br J Dermatol must be included! Also, literature from 2020-2021 of the most established dermatologic journals JAMA Dermatology, EADV, JAAD, British Journal of Dermatology etc. should be checked for important publications in this regard.
  • The discussion should be shortened and focussed. Rather than recapitulating the cases one by one again, the specifities of the cases should be put into perspective in light of available literature. The first sentence of the discussion is very long and incomprehensive. Start with the most important finding of your study.
  • Page 8 Line 190-191, citation needed
  • Please improve the dermatological descriptions both in the figure legends and in the main text; "element" is not a customary term to describe skin lesions
  • The tables include numerous laboratory markers which are normal; focusing on pathological findings would be more appealing to the reader
  • Case II: in the case description it does not state a dentist or an antibiotic, in the discussion it does
  • Figure 5 does not display maculopapular elements. The patient appears to suffer from rosacea, on acral sites, her skin looks almost normal

Minor remarks:

  • normally, there should be no citations in the abstract
  • in the introduction, a brief paragraph about MERS would be adjunctive
  • Case 1: it is unclear why the Girl was admitted to the Hospital, because of the urticarial reaction itself? Figure 1: are all Pictures from the same Point of time?
  • To the best of my knowledge, hemoleukogram is not an English word; biological examination is an uncommon term to describe laboratory investigations
  • 0-15 Age Group: better to say "Children"

Round 2

Reviewer 2 Report

The manuscript has been improved considerably, thank you for your thorough revision.

All Major aspects have been addressed.